



# fair-calibrate v1.4.1: calibration, constraining and validation of the FaIR simple climate model for reliable future climate projections

Chris Smith[1,2], Donald P. Cummins[1], Hege-Beate Fredriksen[3], Zebedee Nicholls[2,4,5],
Malte Meinshausen[4,5], Myles Allen[6], Stuart Jenkins[6], Nicholas Leach[6], Camilla Mathison[1,7], and
Antti-Ilari Partanen[8]

[1]School of Earth and Environment, University of Leeds, Leeds LS2 9JT, United Kingdom
[2]Energy, Climate and Environment Program, International Institute for Applied Systems Analysis (IIASA), 2361 Laxenburg, Austria
[3]UiT the Arctic University of Norway, Tromsø, Norway
[4]School of Geography, Earth and Atmospheric Sciences, The University of Melbourne, Melbourne, Australia
[5]Climate Resource, Melbourne, Victoria, Australia
[6]Atmospheric, Oceanic and Planetary Physics, University of Oxford, Oxford OX1 3PU, United Kingdom
[7]Met Office Hadley Centre, Exeter EX1 3PB, United Kingdom
[8]Climate System Research, Finnish Meteorological Institute, Helsinki, Finland

**Correspondence:** Chris Smith (c.j.smith1@leeds.ac.uk)

**Abstract.** Simple climate models (also known as emulators) have re-emerged as critical tools for analysis of climate policy. Emulators are efficient and highly parameterised, where the parameters are tunable to produce a diversity of global mean surface temperature (GMST) response pathways to a given emissions scenario. Only a small fraction of possible parameter combinations will produce historically consistent climate hindcasts, a necessary condition for trust in future projections.

Alongside historical GMST, additional observed (e.g. ocean heat content) and emergent climate metrics (such as the equilibrium climate sensitivity) can be used as constraints upon the parameter sets used for climate projections. This paper describes a multi-variable constraining package for the FaIR simple climate model (FaIR versions 2.1.0 onwards) using a Bayesian framework. The steps are firstly to generate prior distributions of parameters for FaIR based on Coupled Model Intercomparison Project (CMIP6) Earth System models or Intergovernmental Panel on Climate Change (IPCC) assessed ranges, secondly to

generate a large Monte Carlo prior ensemble of parameters to run FaIR with, and thirdly to produce a posterior set of parameters constrained on several observable and assessed climate metrics. Different calibrations can be produced for different emissions datasets or observed climate constraints, allowing version-controlled and continually updated calibrations to be produced. We show that two very different future projections to a given emission scenario can be obtained using emissions from the IPCC Sixth Assessment Report (AR6) (`fair-calibrate` v1.4.0) and from updated emissions datasets through 2022

(`fair-calibrate` v1.4.1) for similar climate constraints in both cases. `fair-calibrate` can be reconfigured for different source emissions datasets or target climate distributions, and new versions will be produced upon availability of new climate system data.



# 1 Introduction

Simple climate models (also known as emulators) are designed to replicate the large-scale behaviour of more complex Earth
system models. Emulators can be statistically-based such as Gaussian process emulators, or physically-based where the equa-
tions of the model can be written analytically and relationships are based on physical understanding, where possible. The
Finite-amplitude Impulse Response (FaIR) model (Millar et al., 2017; Smith et al., 2018; Leach et al., 2021) and many other
reduced complexity climate models (Nicholls et al., 2020, 2021) are of the latter type. Emulators project mean temperatures for
the whole globe or a few aggregated regions on a monthly or annual timestep, rather than replicating a full 3D atmosphere and
ocean at sub-hourly timesteps such as in Earth System models (ESMs). What emulators lack in spatial, temporal and physical
detail is made up for in efficiency and flexibility. Some emulators may only report GMST as a climatic output. However, several
regional climate variables (Mathison et al., 2023; Wells et al., 2023) and climate impacts (Shiogama et al., 2022) are shown to
scale with GMST, and GMST is often used as a proxy for impacts and damages in climate policy discussions (e.g. the 1.5°C
and 2°C warming levels of the Paris Agreement) and economic models (Howard and Sterner, 2017). Emulators are *efficient*
and may run at tens, hundreds, or thousands of model years per wallclock second, compared to the model years per wallclock
day yardstick for Earth system models. Simple climate models are also *flexible* and highly parameterised, meaning that a wide
range of climate behaviour can be explored by varying parameter choices.

These two features of efficiency and flexibility make it possible to run large probabilistic ensembles using emulators to
explore the range of climate uncertainty to a given emissions scenario. While a number of ESMs exist, allowing us to explore
differences in model responses to forcing, their relatively small number represent an ensemble of opportunity (Tebaldi and
Knutti, 2007) and means that projections using ESMs alone likely under-explores the uncertainty space. It has also been well-
publicised that several CMIP6 models have equilibrium climate sensitivity (ECS) outside of the *very likely* (nominal 5–95%)
range assessed by the IPCC in AR6 (Forster et al., 2021), with other expert assessments coming to similar conclusions about
the range of ECS (Sherwood et al., 2020). Many CMIP6 models show a poor reconstruction of historical temperatures (Smith
and Forster, 2021), with future climate projections run with only a small number of Shared Socioeconomic Pathway (SSP)
scenarios (O'Neill et al., 2016) that start in 2015. These simulations are therefore rapidly becoming outdated, which means
that unadjusted GMST projections from CMIP6 models are often not appropriate for understanding climate change responses
to anthropogenic emissions and assessing impacts of climate policy, particularly on the short timescales that policymakers
need.

Flexibility can be a double-edged sword. Emulators are only useful if the climate projections they provide are reliable. It is
therefore critical that emulators are calibrated to reproduce, at the very least, the time series of historical GMST to a satisfactory
standard. The IPCC AR6 Working Group 1 (WG1) provided a rigorous calibration of four emulators (MAGICC v7.5.3, FaIR
v1.6.2, CICERO-SCM and OSCAR v3.1.1) against historical observations of GMST and ocean heat content (OHC) change and
IPCC-assessed distributions of ECS, transient climate response (TCR), transient climate response to cumulative $CO_2$ emissions
(TCRE), present-day aerosol forcing and future projections of warming under SSP scenarios, including their uncertainties.
Three of the emulators, including FaIR, were assessed to be suitable to be taken forward for use by the IPCC AR6 Working





Group 3 (WG3) to produce warming projections from emissions pathways derived from integrated assessment models (IAMs) (Riahi et al., 2022). Over 1800 scenarios were assessed by WG3, rendering this task impossible for ESMs and necessitating the existence of reliable, well-calibrated emulators.

In this paper we develop and formalise the calibration code for FaIR, developed originally as part of the IPCC AR6 WG1– WG 3 handshake over the course of 2021 and 2022 (Kikstra et al., 2022). The `fair-calibrate` package is available as an open-source Python and R library that builds upon the IPCC AR6 WG1 calibration process for the FaIR model, and designed to work with FaIR model versions starting at v2.1.0, with a future backport to v2.0.0 planned. The versions of `fair-calibrate` described in this paper are run with FaIR v2.1.3. `fair-calibrate` is designed to be flexible, easy to

update, and has a clearly-defined version control strategy. We aim to provide updated constrained probabilistic projections of near-term and 21st Century warming using FaIR at least annually to coincide with the Indicators of Global Climate Change (IGCC) Project (Forster et al., 2023), as new emissions and data for updating observational constraints become available. The headline calibration version in this paper, v1.4.1, is the first example of this, with emissions and observational constraints updated through 2022. For comparison, we also provide an updated IPCC AR6 calibration (v1.4.0), using historical emissions

to 2014 and projections thereafter, showing the significant impact of using different historical emissions datasets for projections.

     Section 2 discusses the code requirements and version control strategy. Section 3 describes the process chain for calibrating FaIR, focusing on `fair-calibrate` v1.4.1. Section 4 shows results of the calibrations v1.4.1 and v1.4.0 compared to IPCC assessed climate indicators and their updates. Section 5 concludes.

## 2    Calibration requirements, versions and versioning strategy

### 70    2.1    Requirements and reproduction

`fair-calibrate` is a collection of Python and R scripts and developed on GitHub, with each version's source code, intermediate data and final output released with digital object identifiers (DOIs) on Zenodo (Smith, 2024). Required dependencies are Python versions 3.8 to 3.11 and R $\geq$ 4.1.1. The `fair-calibrate` package requirements are managed through the Anaconda Python and R package manager, which is also required. `fair-calibrate` sits independently of the FaIR source code

which is deliberately kept clean.

     Each calibration release contains one or more CSV files of parameters and model configuration settings that allow for reproducibility of the calibration of any emissions scenario run in FaIR, and a larger ZIP file containing all results, source files, and intermediate output data produced by the calibration code, so that users can inspect and quickly perform their own analysis on the prior ensemble generated without having to re-run the calibration. The ZIP files also contain diagnostic plots

generated by the code, many of which are included in this manuscript. Intermediate output files and plots are not part of the GitHub repository owing to their file sizes.



## 2.2 Version control strategy

`fair-calibrate` does not strictly adhere to semantic versioning, but sequential version control allows for exact reproducibility and easy comparison of calibrations. As with semantic versioning, the version string is of the form `vX.Y.Z`. Any change in calibration strategy that represents a departure from previous logic would increment the major version `X`, congruent with a "breaking change" in semantic versioning parlance. If an update to an existing calibration or constraining process would change previously submitted results if they were to be re-run with the same emissions and constraints, this is a minor version `Y` increment. Examples of minor version updates include bug fixes and changes in some of the prior distribution ranges used for sampling (section 3.2). The micro version `Z` pertains to either the constraint set or the historical emissions data used. This allows different sets of emissions or constraints to be run with the same overall calibration strategy for easy comparison. Unlike in semantic versioning, an increment of `Z` does not necessarily imply a bug fix or that a more recent version is in some way superior than an older version, or any parallels in the `Z` value between different `vX.Y` since calibrations are developed and released whenever a new use case arises. It is not always possible for different `Z` micro versions to be exactly directly comparable, but the overall *sentiment* should be to change as little as possible other than emissions and/or constraints.

## 2.3 Calibration versions in the v1.4 series

The most recent minor version 1.4 is the focus of this paper. While methods and results presented here are specific to v1.4, this paper is designed to serve as an overall reference to the `fair-calibrate` method and is intended to be a valid guidance document for many future versions.

### 2.3.1 v1.4.1: best estimate historical emissions 1750–2022

`fair-calibrate` v1.4.1 uses up-to-date historical emissions as far as possible and the emissions are as follows:

– $CO_2$ emissions for both fossil fuel & industrial (FFI) and agriculture, forestry & other land-use (AFOLU) $CO_2$ are from the Global Carbon Project 2023 v1.0 (Friedlingstein et al., 2023)

– $CH_4$ and $N_2O$ from non-biomass burning sources, plus $SF_6$, $NF_3$, and aggregated HFCs and PFCs are from PRIMAP-HistTP v2.5 (Gütschow and Pflüger, 2023; Gütschow et al., 2016)

– Short-lived climate forcers, comprising black carbon (BC), organic carbon (OC), sulfur dioxide ($SO_2$), nitrogen oxides (NOx), ammonia ($NH_3$), carbon monoxide (CO) and volatile organic compounds (VOCs), from fossil, industrial and agricultural sources, are from the Community Emission Data System (CEDS) v2021.04.06 (O'Rourke et al., 2021; Hoesly et al., 2018)

– Biomass burning emissions of $CH_4$, $N_2O$ and short-lived climate forcers (SLCFs) are taken from the Global Fire Emissions Database (GFED) (van der Werf et al., 2017) v4.1, which includes the BB4CMIP dataset prepared for CMIP6 historical simulations (van Marle et al., 2017)





- Emissions of Montreal Protocol greenhouse gases (CFCs, HCFCs, Halons, chlorinated and brominated gases), along with $SO_2F_2$, are estimated using inverse greenhouse gas concentrations that have been prepared for the IGCC (Forster et al., 2023), as no inventories of these emissions datasets are available to our knowledge.

All emissions datasets are produced for 1750–2022, except CEDS which has a 2019 end date. To extend SLCFs from CEDS to 2022, we use the "two year blip" scenario that estimates the decline and recovery from emissions due to COVID-19 from Forster et al. (2020) and extended by Lamboll et al. (2021), based on proxy activity data. We take the ratios of SLCF emissions species over 2020–2022 to 2019 in the two-year blip scenario, and apply them as a scaling factor to CEDS emissions in 2019. Such a version-controlled strategy allows for the calibration to be updated as newer emissions data becomes available. Emissions data prepared to the end of 2023 will be available over the course of 2024, and an anticipated update to CEDS should also bring non-biomass burning SLCFs until at least the end of 2022 (Hoesly et al., 2023). This demonstrates that "operational" calibrations are often a moving target.

We use the "third-party" emissions from PRIMAP-Hist (the PRIMAP-HistTP dataset) rather than country-reported (PRIMAP-HistCR) values, on the assumption that we expect solely country-reported values to be an underestimate of true emissions. We demonstrate that HistTP still appears to be an underestimate for many species based on best estimate greenhouse gas lifetimes and concentration estimates.

### 2.3.2 v1.4.0: RCMIP historical emissions prepared for AR6 (1750–2014)

For consistency and comparison with the FaIR projections used in the IPCC AR6, we produce a calibration using historical emissions from RCMIP (Nicholls et al., 2020, 2021) using v5.1.0 of the Reduced Complexity Model Intercomparison Project (RCMIP) emissions dataset available from Nicholls and Lewis (2021). The RCMIP emissions contain global annual total emissions of $CO_2$ and SLCFs that were prepared for running CMIP6 models. Emissions of non-$CO_2$ greenhouse gases were back-calculated to reproduce the CMIP6 best-estimate historical concentrations (Meinshausen et al., 2017). These concentrations time series were also used to drive CMIP6 models.

For SSP scenarios, emissions from 2015 to 2100 were produced using IAMs, which were then extended to 2500 using simplified assumptions (Meinshausen et al., 2020). We use the same climate constraints on GMST, $CO_2$ concentration and OHC as for v1.4.1 (section 3.3), datasets which run to 2022. For the bridging period 2015–2022 between the end of the CMIP6 historical and the observational climate data, we use emissions from SSP2-4.5, expected to be the closest Tier 1 SSP to current policies (Hausfather and Peters, 2020), and as shown later, the closest Tier 1 scenario to post-2015 emissions.

One adjustment is made to the RCMIP emissions to correct NOx. For accounting purposes we express NOx in units of $Tg \ NO_2 \ yr^{-1}$. The source datasets for RCMIP were earlier versions of CEDS, which reports emissions in $Tg \ NO_2 \ yr^{-1}$ for fossil-fuel and agricultural emissions, and GFED, which reports emissions in $Tg \ NO \ yr^{-1}$ for biomass burning. The conversion for GFED emissions data was not made in RCMIP v5.1.0.

Neither `fair-calibrate` v1.4.1 nor v1.4.0 includes forcing from aviation contrails. Forcing from contrails and its temperature impact were assessed in the IPCC AR6 WG1 (Forster et al., 2021), with best estimate contributions to present-day





forcing of 0.06 W m$^{-2}$ and warming of 0.02°C, and included in the WG1 calibration of FaIR. However, contrail forcing was excluded from the WG3 IAM emissions projections, rendering the WG1 and WG3 projection sets slightly inconsistent. To project contrails forcing into the future requires estimates of aviation activity. FaIR can accept a time series of contrails forcing directly, or estimate it using a linear combination of emissions species. By default, FaIR uses NOx emissions from the aviation sector to estimate contrail forcing (Smith et al., 2018). Neither aviation activity nor NOx emissions from aviation are provided

in IAM scenarios in general, so contrails forcing could not be assessed in WG3. Aviation NOx emissions are provided in the RCMIP historical and SSP future emissions and could be included in `fair-calibrate` v1.4.0. However, in order to apply the calibrations consistently to as many scenarios as possible, we calibrate without them.

## 3 Process

The set of output FaIR parameters is produced in three steps: (1) calibration; (2) sampling; and (3) constraining. The description

and results in this section apply generally to all calibration versions to date. We focus on calibration v1.4.1, and describe methods pertinent to v1.4.0 where they differ.

### 3.1 Calibration

#### 3.1.1 Climate response

The climate response module of FaIR v2.1.3 is an impulse-response formulation of the three-layer stochastic energy balance

model of Cummins et al. (2020). We calibrate this model using 150-year $4\times CO_2$ experiments from 49 CMIP6 models, using GMST ($\Delta T_1$) and top of atmosphere energy imbalance ($\Delta N$) as anomalies relative to each model's pre-industrial control run, subtracting a linear trend from the appropriate branch point of each model's control to account for any residual drift. This calibration is performed using the maximum-likelihood method of Cummins et al. (2020), and the `EBM` R package that accompanies Cummins et al. (2020) is used in the `fair-calibrate` process chain (Cummins, 2021).

The three-layer stochastic energy balance model is written as

$$C_1 \frac{dT_1(t)}{dt} = F(t) - \kappa_1 T_1(t) - \kappa_2 (T_1(t) - T_2(t)) + \xi(t) \tag{1}$$

$$C_2 \frac{dT_2(t)}{dt} = \kappa_2 (T_1(t) - T_2(t)) - \varepsilon \kappa_3 (T_2(t) - T_3(t)) \tag{2}$$

$$C_3 \frac{dT_3(t)}{dt} = \kappa_3 (T_2(t) - T_3(t)). \tag{3}$$

In eqs. (1) to (3), $T_1$, $T_2$ and $T_3$ are the temperature anomalies of the three ocean layers (starting from the surface), $C_1$, $C_2$ and

$C_3$ are their heat capacities, $\kappa_j$ represents the heat transfer coefficients between layers $j-1$ and $j$ for $j \geq 2$, $-\kappa_1$ is the climate feedback parameter (often denoted $\lambda$), $\varepsilon$ is the deep ocean efficacy parameter (Held et al., 2010; Winton et al., 2010; Geoffroy et al., 2013), $\xi$ is a stochastic disturbance term in the temperature response that does not affect the top-of-atmosphere energy imbalance, and $F$ is the effective radiative forcing (ERF).





The effective radiative forcing is the sum of a deterministic and stochastic component $F = F_{\text{det}} + \zeta$. The stochastic forcing

component $\zeta$ is modelled as a continuous-time red-noise process

$$\frac{d\zeta}{dt} = -\gamma\zeta + \eta \tag{4}$$

where $\eta$ is white noise and $\gamma > 0$ controls the strength of temporal auto-correlation (Cummins et al., 2020). In FaIR, the stochastic behaviour can be switched off, and eqs. (1) to (4) reduce to a deterministic energy balance model when $\xi = \eta = 0$ (Geoffroy et al., 2013; Leach et al., 2021).

The top-of-atmosphere energy imbalance $N$ is given as

$$N(t) = F(t) - \kappa_1 T_1(t) + (1-\varepsilon)\kappa_3(T_2(t) - T_3(t)) \tag{5}$$

and the Earth's energy uptake, used as a model constraint, is the time integral of $N$.

For each of the 49 CMIP6 models, we obtain a set of 11 parameters $\{C_1, C_2, C_3, \kappa_1, \kappa_2, \kappa_3, \varepsilon, \gamma, \sigma_\xi, \sigma_\eta, F_{4\times CO_2}\}$ that describes the magnitude and rate of warming to a $4\times CO_2$ forcing (fig. S1), and the behaviour of internal variability where $\sigma_\xi$

and $\sigma_\eta$ are the standard deviations of $\xi$ and $\eta$ around zero mean. $F_{4\times CO_2}$ is the effective radiative forcing from a quadrupling of pre-industrial $CO_2$ concentrations. The comparison of one stochastic realisation of each model's energy balance model calibration (black) compared to the actual CMIP6 model (red) for the temperature response to an abrupt $4\times CO_2$ forcing is shown in fig. 1. In almost all cases the FaIR calibration is an excellent representation of the underlying CMIP6 model. The calibrated parameters are shown in table S1.

The energy balance model parameters can be written as a matrix equation that describes the time evolution of each temperature layer (Cummins et al., 2020; Leach et al., 2021). The impulse-response form of the temperature evolution in each layer can be calculated from the eigenvalues and eigenvectors of the energy balance matrix. From this, the ECS and "theoretical" TCR for each model calibration can be directly estimated from the impulse-response coefficients as described in Leach et al. (2021, section 2.4). The ECS calculated here is a true equilibrium value rather than as a regression over a 150-year simulation

as usually performed from ESM output (the so-called effective sensitivity, EffCS). The theoretical TCR is not precisely what each model would predict after 70 years of a 1% compound increase of atmospheric $CO_2$ concentrations, but is usually close and has the advantage that model simulations do not need to be run to determine this value (fig. S1b).

### 3.1.2   Minor greenhouse gas emissions

This section describes the emissions adjustment procedure in `fair-calibrate` v1.4.1 for emissions of minor greenhouse

gases. In this context "minor" means any species that is not $CO_2$ or $CH_4$. This includes $N_2O$, hydroflouorocarbons (HFCs), perfluorocarbons (PFCs), $SF_6$ and $NF_3$. This emissions adjustment is not required in v1.4.0 where emissions from all species are provided by the RCMIP emissions datasets (section 2.3.2).

HFCs and PFCs are provided in PRIMAP-Hist as aggregate values reported in $CO_2$-equivalent (AR6 $GWP_{100}$) emissions. We disaggregate these emissions by scaling the annual historical emissions totals in $CO_2$-eq from RCMIP historical + SSP2-4.5

for 1750–2022 to the PRIMAP-Hist reported values, then multiplying this scaling by the RCMIP individual species emissions value in each year. Table S2 details the HFC and PFC gases included in the disaggregation.





**Figure 1.** Comparison of temperature projections from abrupt $4 \times CO_2$ simulations as calibrated in FaIR (black) to the original CMIP6 model results (red) for 49 CMIP6 models. For FaIR we show one realisation with stochastic internal variability included; different random seeds would produce different internal variability profiles.





The following step calculates atmospheric concentrations when run forward using a single time-constant decay model with the PRIMAP-Hist emissions and time constants equal to atmospheric lifetimes assessed in IPCC AR6 (Smith et al., 2021a). The calculated concentration time series is compared to the best-estimate historical concentrations from Forster et al. (2023),

which is an update of the AR6 concentrations in IPCC (2021) to 2022 using recent AGAGE and NOAA station data. In many cases, the calculated and observed concentrations differ substantially, and the calculated concentrations are usually lower than the observed. This implies that either the reported emissions in PRIMAP-Hist do not capture all true emissions, or that the reported atmospheric lifetimes are too short (a third, less likely possibility is that the reported concentrations are too high). A correction can be obtained by either lengthening the lifetimes or scaling up the emissions. We choose to adjust the emissions

on the basis that countries under-reporting due to incomplete data is plausible, and scaling the emissions brings some species much closer to RCMIP estimates which are derived from inverting atmospheric concentrations. The scaling is performed in order to match the projected concentrations to the historical best estimates in 2019. In many cases the scaling is mild (for $N_2O$, emissions are scaled up by a factor of 1.08; fig. 2a) but can be large ($NF_3$ is scaled by a factor of 7.5; fig. S2). This implies that countries are severely under-reporting emissions of some GHGs compared to the increasing stock of these gases observed in

the atmosphere.

PRIMAP-Hist does not provide emissions of $SO_2F_2$ or of Montreal protocol GHGs. We estimate their emissions by inverting the concentrations time series in Forster et al. (2023).

For future projections, we harmonize to 2022 (Gidden et al., 2018) the eight Tier 1 and Tier 2 SSP scenarios to our scaled calculated historical emissions. This produces SSPs that take into account the recent past. We can then compare the harmonized,

adjusted future concentrations projections to those created for the SSP scenarios that used MAGICC6 (Meinshausen et al., 2020). Figure 2b shows recreated historical and future $N_2O$ concentration projections to 2100 under eight SSP scenarios using the harmonized, scaled emissions (thick lines) in FaIR and their comparison to the SSP concentrations time series (thin lines) from Meinshausen et al. (2017, 2020). Note that the historical concentrations differ between (a) and (b) as the dataset sources differ. For $N_2O$ the correspondence between FaIR and CMIP6 is very good for all eight SSPs for future projections.

### 3.1.3  Methane lifetime

A new feature of FaIR introduced in v2.1.0 is a variable methane lifetime that depends on burdens of chemically reactive species and climate. This is an update from v2.0.0 that used a methane lifetime self-feedback (methane concentrations and temperature affects climate) and previous versions that did not modify the lifetime of methane at all.

A methane lifetime scaling factor $\alpha_{CH_4}$ is applied to the base lifetime $\tau_{CH_4,base}$ calculated as

$$\log \alpha_{CH_4} = \log(1 + S_T \Delta T_1) + \sum_i \log(1 + S_i \Delta A_i). \tag{6}$$

In eq. (6), $S_i$ denotes a sensitivity to species $i$ or GMST anomaly ($\Delta T_1$) and $\Delta A_i$ represents abundances of species $i$ (emissions rate for SLCFs and concentrations for GHGs) of chemically reactive species. If the anomalies in temperature and abundances are relative to pre-industrial, $\alpha_{CH_4} = 1$ in pre-industrial conditions and $\tau_{CH_4,base}$ is pre-industrial lifetime.



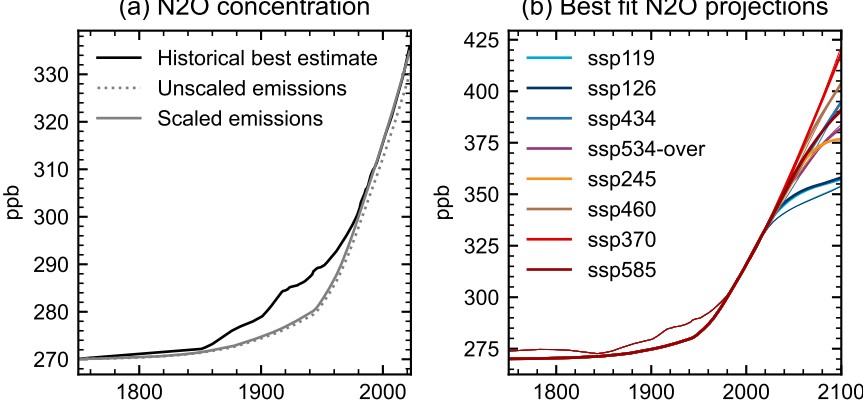

**Figure 2.** (a) Comparison of best estimate historical N$_2$O emissions (black), the concentration projected from emissions in PRIMAP-Hist + GFED (grey dotted) and the concentrations after scaling up the emissions by a factor of 1.08 to get correct recent historical concentrations (grey solid). Note that a single lifetime cannot accurately reproduce best estimate historical concentrations between 1850 and 1950. (b) Harmonized SSP projections using the scaled historical emissions (thick lines), compared to the SSP historical + future projections (thin lines) from Meinshausen et al. (2017, 2020).

Unlike for minor GHGs, emissions are not scaled for CH$_4$ in `fair-calibrate` v1.4.1, and we instead calibrate the
atmospheric chemical lifetime. Owing to dependence of the lifetime of several simultaneously changing emissions species as well as climate, there is not a unique invertible concentration to emissions pathway for methane.

The UKESM1.0-LL, GFDL-ESM4, GISS-E2.1-G and MRI-ESM2.0 Earth System models provide a complete set of results from the Aerosol Chemistry Model Intercomparison Project (AerChemMIP) single-forcing experiments that enable estimation of the sensitivity in methane lifetime to climate (Thornhill et al., 2021a) and chemically reactive species (Thornhill et al.,
2021b). We use results reported in Thornhill et al. (2021b, a) for methane lifetime in 1850 and its relative sensitivity to changes in CH$_4$, N$_2$O and equivalent effective stratospheric chlorine (EESC) concentration, emissions of NOx and VOCs, and global mean surface temperature between 1850 and 2014 in each of the four models. For each atmospheric species, the fractional change in lifetime in 2014 relative to 1850 is normalized by the burden change, to provide lifetime changes in each model in terms of parts per billion concentration change or Mt yr$^{-1}$ emissions. The four models that provide data are used as minimum
and maximum ranges of a parameter search (in v1.4.1, we expand the search range by a factor of two, since the PRIMAP-Hist methane emissions are again likely to be an underestimate and do not find suitable parameters within the model range) to minimize the difference between observed CH$_4$ concentrations from Forster et al. (2023) and those calculated from eq. (6). 1750 emissions are subtracted from the time series when performing the lifetime calibration as it is assumed that pre-industrial concentrations of methane are in approximate equilibrium with pre-industrial concentrations.
The historical best estimate calibrations are shown in table 1. It can be seen that the methane lifetime in `fair-calibrate` v1.4.1 is nearly 17 years in the pre-industrial period, which is much longer than typically determined from ESMs. The best



estimate lifetime in FaIR from historical emissions is shown in fig. 3a (grey line), and is indeed longer than that calculated from the sensitivities in each CMIP6 model across most of the historical period, though close to the AR6 value in the present day. In fig. 3b the historical concentrations from Forster et al. (2023) (black) are compared to the best estimate from FaIR using the lifetime calculated in (a) and run forward with best estimate historical emissions. In fig. 3c, the SSP methane concentrations are projected with the harmonized emissions starting in 2022, and compared to the SSP concentrations time series (Meinshausen et al., 2017, 2020). In general, the harmonized methane concentration projections from `fair-calibrate` v1.4.1 are lower than in CMIP6 for high methane emissions scenarios, and higher for low emissions futures. This is due in part to the nearly 10 years of additional historical emissions in the best estimate time series compared to the SSPs, which started to diverge from a common history in 2015. For these projections we use the best estimate GMST anomalies from the SSPs derived in Lee et al. (2021).

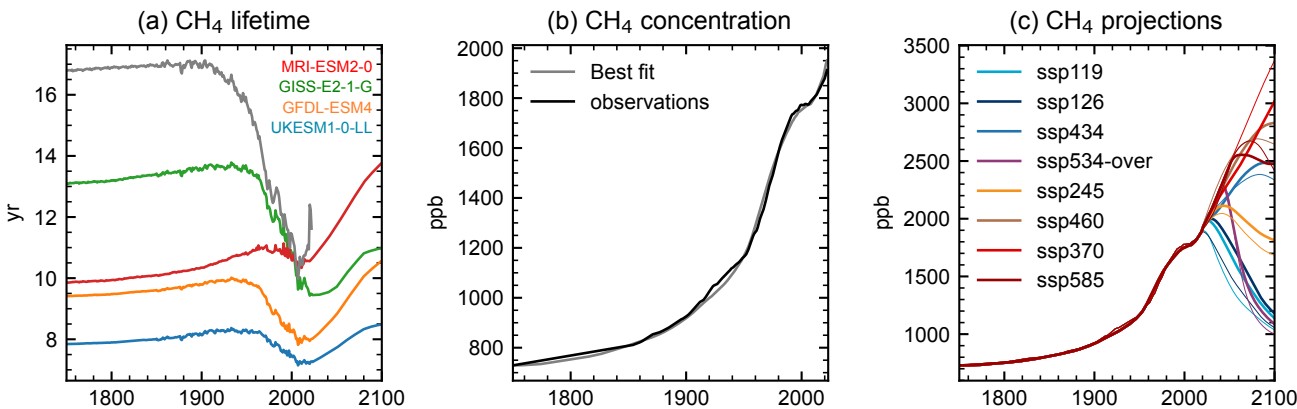

**Figure 3.** Methane lifetime calibration (v1.4.1). (a) Methane lifetime in the historical+SSP3-7.0 scenario for four ESMs (colours) and the lifetime from the FaIR calibration (grey). (b) Methane concentration calculated from historical methane emissions from PRIMAP-Hist + biomass burning emissions using the lifetime in (a), using FaIR (grey), and the observed atmospheric concentrations (black) for 1750–2022 from IGCC (Forster et al., 2023). (c) Methane concentrations calculated from methane emissions for the eight main SSP scenarios using the harmonized future emissions projections (thick lines) compared to the SSP scenarios (thin lines) (Meinshausen et al., 2017, 2020).

The lifetimes, historical and future calibrations for the RCMIP emissions (calibration v1.4.0) are shown in fig. S4, where it is observed that lifetimes and concentration projections are much closer to AR6 and CMIP6. This demonstrates that firstly the calibration is plausible (CMIP6 emissions give CMIP6 concentrations), and secondly that the methane lifetime calibration is very sensitive to the historical emissions time series used. In fig. 4b we compare the methane emissions from the v1.4.0 and v1.4.1 calibrations. As 1750 emissions are subtracted from the total to report changes away from a pre-industrial equilibrium, the change in emissions (1750–2022) in v1.4.1 from PRIMAP-Hist is smaller than in v1.4.0, leading to longer atmospheric lifetimes necessary to reproduce concentrations.



| Variable | Best historical fit v1.4.1 | Best historical fit v1.4.0 |
|---|---|---|
| Lifetime in 1750 | 16.8 yr | 10.0 yr |
| $CH_4$ sensitivity | $1.67 \times 10^{-4}$ ppb$^{-1}$ | $2.54 \times 10^{-4}$ ppb$^{-1}$ |
| $N_2O$ sensitivity | $-9.50 \times 10^{-4}$ ppb$^{-1}$ | $-7.23 \times 10^{-4}$ ppb$^{-1}$ |
| EESC sensitivity | $2.53 \times 10^{-5}$ ppt$^{-1}$ | $-5.33 \times 10^{-6}$ ppt$^{-1}$ |
| NOx sensitivity | $-3.42 \times 10^{-3}$ (MtNO$_2$ yr$^{-1}$)$^{-1}$ | $-2.52 \times 10^{-3}$ (MtNO$_2$ yr$^{-1}$)$^{-1}$ |
| VOC sensitivity | $1.98 \times 10^{-3}$ (MtVOC yr$^{-1}$)$^{-1}$ | $1.62 \times 10^{-3}$ (MtVOC yr$^{-1}$)$^{-1}$ |
| temperature sensitivity | $-0.0463$ K$^{-1}$ | $-0.0408$ K$^{-1}$ |

**Table 1.** Baseline $CH_4$ lifetime, and sensitives ($S_i$) in lifetime due to changes in greenhouse gas concentrations, short-lived climate forcer emissions and temperature in calibrations v1.4.1 and v1.4.0.

Unlike in versions of FaIR prior to 2.0.0, we do not assume any natural methane emissions. In v1.3 of FaIR for example, natural emissions were back-calculated with the assumption of a constant methane lifetime and held constant for future projections (Smith et al., 2018). It is well-known that wetlands emit large quantities of methane, and it is very likely that this effect is climate-dependent (Zhang et al., 2017). As the climate continues to warm, biogenic methane will be released from permafrost soils and clathrates—sources that most ESMs do not include at present. Including these natural sources is a development priority for future versions of FaIR.

### 3.1.4 Carbon cycle feedbacks

The carbon cycle is parameterised as a simple atmospheric decay model with four time constants, based on the impulse-response functions of Joos et al. (2013). The time constants are scaled by a lifetime scaling factor that mimics the influence of carbon cycle feedbacks. This treatment is unchanged since Leach et al. (2021, section 2.1). A positive carbon cycle feedback reduces the efficacy of carbon sinks, thus effectively lengthening the atmospheric lifetime of $CO_2$.

The lifetime scaling factor is a function of the time-integrated airborne fraction of a $CO_2$ pulse over 100 years $I_{100}$ (Millar et al., 2017). $I_{100}$ is modified as

$$I_{100} = r_0 + r_U \Delta C_U + r_T \Delta T + r_A \Delta C_A \tag{7}$$

where $r_0$, $r_U$, $r_T$ and $r_A$ are pre-industrial time-integrated airborne fraction and its sensitivity to cumulative carbon uptake in land and ocean sinks $\Delta C_U$, surface temperature anomaly $\Delta T$ and airborne carbon $\Delta C_A$ respectively. Total cumulative emissions since pre-industrial is $\Delta C_A + \Delta C_U$.

The process for calibrating the carbon cycle feedbacks to 11 CMIP6 ESMs containing interactive carbon cycles is described in Leach et al. (2021, section 3.2). The same coefficients derived in Leach et al. (2021) for the 11 ESMs are used in all calibrations to date.





### 3.1.5 Aerosol-cloud interactions

The effective radiative forcing due to aerosol cloud interactions $\mathrm{ERF_{aci}}$ has been generalised:

$$\mathrm{ERF_{aci}} = \beta \left[ \log\left(1 + \sum_i s_i A_i\right) - \log\left(1 + \sum_i s_i A_{i,\mathrm{base}}\right) \right] \tag{8}$$

where $A_i$ is the emissions or concentration of a species and the base subscript denotes its reference (usually pre-industrial) abundance, $\beta$ is a scale factor and $s_i$ describes how sensitive a species is in contributing to ERFaci. The generalisation allows for inclusion of more species that affect ERFaci in addition to $SO_2$, BC and OC that was modelled previously. The generalisation

is useful as there is evidence for a large ERFaci response to $CH_4$ in UKESM1-0-LL through methane's effect on competing for atmospheric oxidants including OH, affecting the rate of new particle formation (O'Connor et al., 2022). As with earlier versions of FaIR, the form of eq. (8) is inspired by Stevens (2015), but without any physical significance attached to the sensitivities $s_i$, allowing near-linear global mean responses in ERFaci to changes in precursor abundances as postulated by some authors (Booth et al., 2018; Kretzschmar et al., 2017) and exhibited in some models (Smith et al., 2021b).

Thirteen CMIP6 models provided results from transient aerosol experiments in AerChemMIP and RFMIP (table 2) that allow calculation of aerosol ERF. The breakdown of shortwave aerosol ERF into aerosol-radiation interactions (ERFari) and ERFaci is performed using the Approximate Partial Radiative Perturbation (APRP) method (Taylor et al., 2007) following the logic of Zelinka et al. (2014) and Zelinka et al. (2023). Longwave contributions to ERFaci are estimated from the cloud radiative effect, with ERFari estimated as the difference between the longwave components of ERF and ERFaci.

From the diagnosed ERFaci in each model, a least-squares curve fit of ERFaci to historical emissions by fitting $s_{SO_2}$, $s_{BC}$, $s_{OC}$ and $\beta$ is found (table 2) using eq. (8). The comparison of model-derived ERFaci to the best fit from eq. (8) is shown in fig. 4.

Using eq. (8), a wide range of ERFaci trajectories are possible, and parameter estimates for $\beta$ and individual species sensitivities span orders of magnitude. Where one or two of $s_{SO_2}$, $s_{BC}$, $s_{OC}$ are close to zero (CanESM5, UKESM1-0-LL), this

indicates that the species has little influence on ERFaci in that model (e.g. UKESM1-0-LL's ERFaci response is purely driven by sulfate in aerosol-only forcing experiments). Where all three of $s_{SO_2}$, $s_{BC}$ and $s_{OC}$ are close to zero and $\beta$ has large magnitude (the two GFDL models, NorESM2-LM), this indicates that ERFaci behaves linearly in emissions, from the Taylor expansion of $\log(1+x)$ for small $x$ (Smith et al., 2021b). In the case of NorESM2-LM, the coefficient for BC is so small that it is effectively zero, with the ERFaci response being linear with sulfate and OC.

### 3.1.6 Ozone


The best estimate historical ozone ERF time series from Skeie et al. (2020) is used to calibrate the role of ozone precursors to ozone forcing. As in AR6, tropospheric and stratospheric ozone are not considered separately. Again following the AR6 methodology, we select six models from the twelve coupled historical CMIP6 models analysed in Skeie et al. (2020) that are relatively independent from each other, have full stratospheric and tropospheric chemistry enabled, and reproduce expected

behaviour for the overall time history of ozone ERF. The six models used are BCC-ESM1, CESM2(WACCM6), GFDL-ESM4,





| Model | CMIP6 protocol | $\beta$ | $s_{SO_2}$ [(MtSO$_2$ yr$^{-1}$)$^{-1}$] | $s_{BC}$ [(MtBC yr$^{-1}$)$^{-1}$] | $s_{OC}$ [(MtOC yr$^{-1}$)$^{-1}$] |
|---|---|---|---|---|---|
| CanESM5 | RFMIP | $-0.856$ | 0.0199 | 0.394 | $1.25 \times 10^{-16}$ |
| CNRM-CM6-1 | RFMIP | $-1.50$ | 0.00601 | 0.0460 | 0.0111 |
| E3SM-2-0 | RFMIP | $-1.44$ | 0.0715 | $1.29 \times 10^{-41}$ | 0.352 |
| GFDL-CM4 | RFMIP | $-4507$ | $1.10 \times 10^{-6}$ | $5.94 \times 10^{-7}$ | $2.13 \times 10^{-6}$ |
| GFDL-ESM4 | AerChemMIP | $-13202$ | $2.54 \times 10^{-7}$ | $2.70 \times 10^{-6}$ | $6.07 \times 10^{-7}$ |
| GISS-E2-1-G | RFMIP | $-0.585$ | 0.00819 | 1.28 | $5.36 \times 10^{-11}$ |
| HadGEM3-GC31-LL | RFMIP | $-0.941$ | 0.0222 | $4.81 \times 10^{-33}$ | 0.0367 |
| IPSL-CM6A-LR | RFMIP | $-1.26$ | 0.00266 | $1.76 \times 10^{-16}$ | 0.00190 |
| MIROC6 | RFMIP | $-1.03$ | 0.00730 | 0.149 | $6.27 \times 10^{-18}$ |
| MPI-ESM-1-2-HAM | AerChemMIP | $-2.35$ | 0.00718 | $3.85 \times 10^{-13}$ | 0.00975 |
| MRI-ESM2-0 | AerChemMIP | $-7.74$ | 0.000776 | 0.00412 | $5.27 \times 10^{-27}$ |
| NorESM2-LM | RFMIP | $-12527$ | $6.91 \times 10^{-7}$ | $2.78 \times 10^{-114}$ | $1.62 \times 10^{-6}$ |
| UKESM1-0-LL | AerChemMIP | $-0.723$ | 0.0335 | $8.76 \times 10^{-37}$ | $6.38 \times 10^{-13}$ |

**Table 2.** Models used to calibrate forcing from aerosol-cloud radiation interactions, and their parameter best fit values from eq. (8).

GISS-E2-1-H, MRI-ESM2-0 and OsloCTM3. Skeie et al. (2020) provides historical ozone forcing for 1850–2010 in these models, and following Skeie et al. (2020) we add +0.03 W m$^{-2}$ to the timeseries to represent the change from 1750 to 1850. The Oslo-CTM3 model provided results under SSP2-4.5 to 2020, which was also used in calibration.

As ozone ERF includes a contribution from temperature change and is calibrated from coupled historical runs, historical
warming is backed out using a temperature feedback of $-0.037$ W m$^{-2}$ K$^{-1}$ (Thornhill et al., 2021a) and historical GMST from Forster et al. (2023). To this "no-feedback" ERF time series, we find a least-squares fit to the change in emissions of NOx, VOC and CO, and concentrations of CH$_4$, N$_2$O and EESC (fig. 5). The lower and upper bounds of the search ranges for the parameter fits are the *very likely* range for each precursor in Thornhill et al. (2021b), which is also scaled up to account for the difference in best estimate ozone forcing between models participating in AerChemMIP in Thornhill et al. (2021b) and the
six-model subset in Skeie et al. (2020).

Similarly to the methane lifetime calibration, we derive a coefficient for each percursor species relating emissions or concentrations of each to the ozone ERF. Uncertainty sampling for the prior distribution is described in section 3.2.5.

## 3.2 Sampling

We produce a 1.6 million member prior ensemble of FaIR projections, with parameter choices drawn from probability distri-
butions that are informed by CMIP6 model calibrations (section 3.1) or AR6 assessed ranges. Different components of FaIR are sampled independently, but within each component (e.g. climate response) the correlation structure between parameters is maintained to ensure internally consistent parameter choices. In many cases, probability distributions for parameters are con-



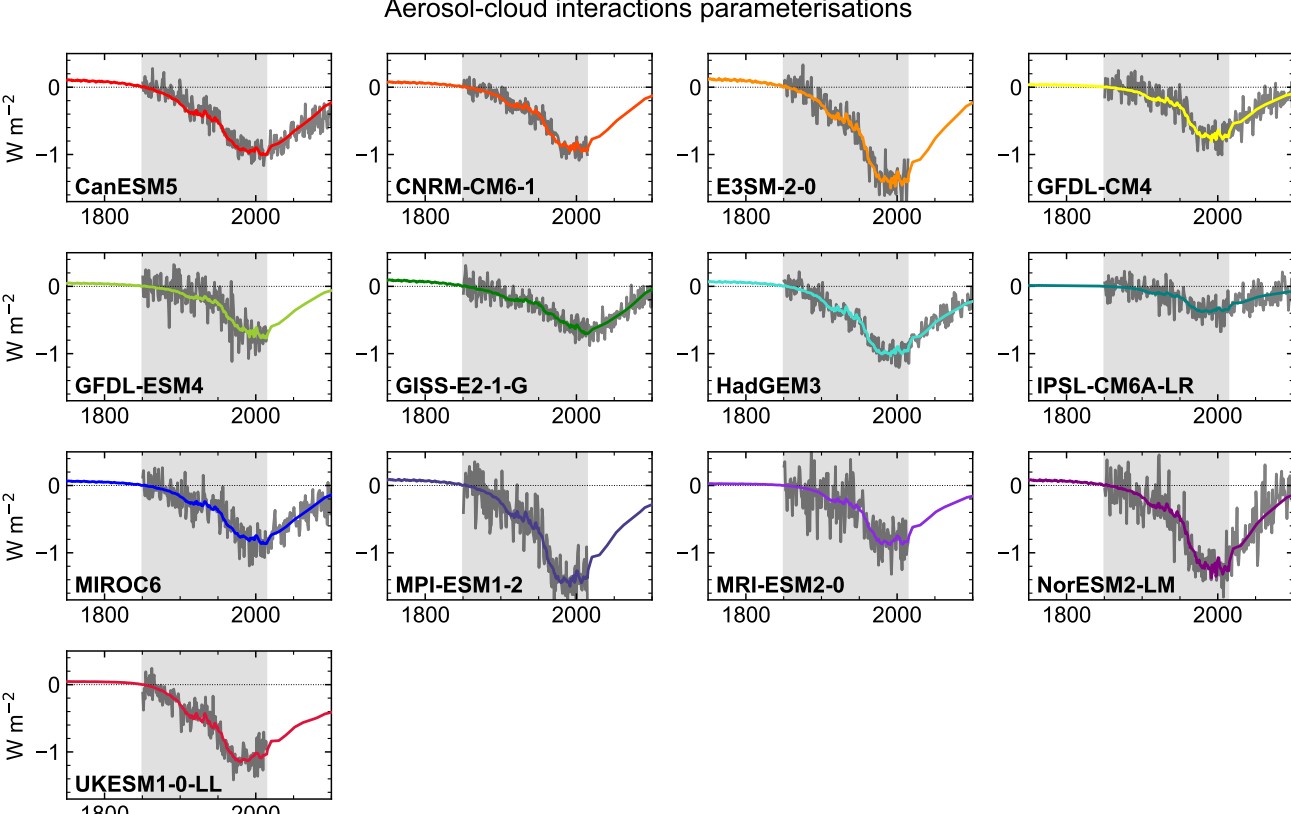

**Figure 4.** Calibrations of the ERFaci relationship in FaIR (eq. (8); coloured lines) to the derived ERFaci from 13 CMIP6 models (grey lines). Extrapolation back to 1750 is shown in all cases, and forward to 2100 under SSP2-4.5 emissions where model simulations were not extended beyond 2014.

structed from a Gaussian kernel density estimate, which is a non-parametric method that attempts to estimate the underlying probability density function from a finite sample size, and can be used to preserve correlation structure in multi-variate cases

(Scott, 1992).

In total, 45 parameters are sampled. In the processing chain, fixed random seeds are used to ensure reproducibility. Internal variability is switched on, and again each parameter set has a random seed associated with it in order to reproduce the same pattern, and key climate metrics are saved out of the prior ensemble.

### 3.2.1 Climate response

An 11-dimensional kernel density estimate is generated from the energy balance model parameters that were calibrated on 49 CMIP6 models (fig. S5). $F_{4 \times CO_2}$ is not used in the climate response of FaIR but is used in the theoretical calculation of

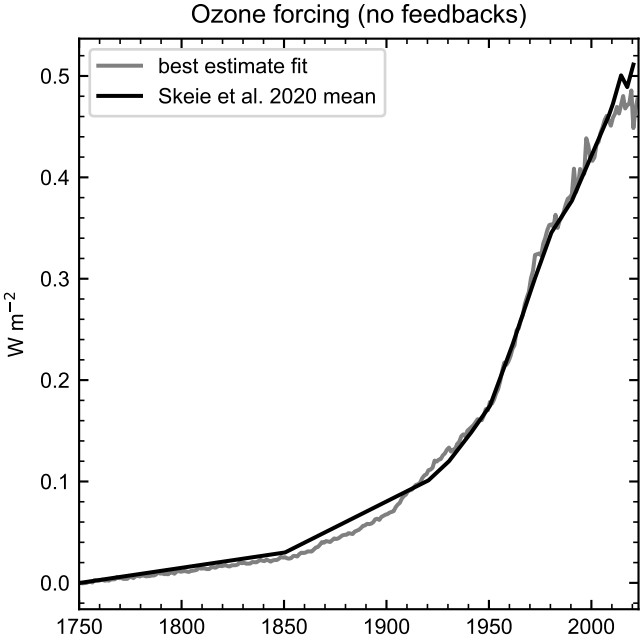

**Figure 5.** Comparison of the ozone ERF timeseries from Skeie et al. (2020) (black) to the estimate from emissions and concentration precursors (grey). The estimated impact of temperature on ozone forcing has been backed out of the time series from Skeie et al. (2020) and is not included in the model fit.

ECS and TCR. Including the correlation between parameters reduces (though does not eliminate) the likelihood of physically implausible combinations being sampled, and using kernel density estimates rather than a parametric multivariate distribution (such as a Gaussian) allows for variability in the distribution shapes of each parameter such as the skewness present in $\kappa_2$
(fig. S5). All parameters of the energy balance model are strictly positive, so parameter sets containing negative values are discarded and redrawn until the 1.6 million threshold is reached. We also discard and redraw instances of $\kappa_1 < 0.3$ W m$^{-2}$ K$^{-1}$, $C_1 < 1.8$ W yr m$^{-2}$ K$^{-1}$, $C_3 < C_2$, $C_2 < C_1$ and $\gamma < 0.5$. The $\kappa_1$ threshold puts an upper bound on the ECS prior of around 13°C, and the other limits ensure model stability.

### 3.2.2   Aerosol-cloud interactions

Similarly to the climate response, we draw correlated kernel density estimates for $\log(s_{SO_2})$, $\log(s_{BC})$ and $\log(s_{OC})$. We calculate an unscaled ERFaci for the 2005–2014 mean relative to 1750 for each parameter set. The unscaled ERFaci is then scaled to reproduce a draw from a trapezoid distribution with limits at $-2.2$ and $+0.2$ W m$^{-2}$ and plateau from $-1.6$ to $-0.4$ W m$^{-2}$ to represent the ERFaci for 2005-2014 relative to 1750, which selects the $\beta$ value to use for that parameter set. This process is similar to that of both Smith et al. (2021b) and AR6 (Forster et al., 2021). The prior distribution is chosen to give a



wide but plausible range around the ERFaci distribution for the present day assessed by the IPCC (Forster et al., 2021), which was $-1.0$ W m$^{-2}$ for a nominal 2014 date relative to 1750.

### 3.2.3    Aerosol-radiation interactions

The ERFari contributions are not sampled directly from CMIP6 models, though much of the basis of this assessment is rooted in AerChemMIP (Thornhill et al., 2021b). AR6 assessed that several species (CH$_4$, N$_2$O, halogenated compounds, sulfate, BC,

OC, nitrate, VOCs) contribute directly or indirectly to ERFari, though only sulfate, BC, OC and NH$_3$ are significant. We use the contributions to ERFari assessed in AR6 with the relative uncertainty from each precursor (Szopa et al., 2021), and scale both the best estimate and uncertainty range of the ERFari from each precursor to reproduce the IPCC AR6 distribution of $-0.3 \pm 0.3$ W m$^{-2}$ (Forster et al., 2021). All ranges quoted are for 5th to 95th percentile unless otherwise stated.

### 3.2.4    Carbon cycle and initial CO$_2$ concentration

A four-dimensional kernel density estimate is drawn from the $r_0$, $r_U$, $r_T$ and $r_A$ parameters from the 11 models calibrated in Leach et al. (2021). As part of the carbon cycle sampling, we draw CO$_2$ concentration values in 1750 using the IPCC AR6 best estimate and uncertainty of $278.3 \pm 2.9$ ppm (5–95%) range (Gulev et al., 2021) using a Gaussian distribution.

### 3.2.5    Ozone

The coefficients relating emissions or concentrations of chemically-relevant precursors to ozone ERF take their mean value

from the bounded least-squares fit derived in section 3.1.6, and their uncertainty values are sampled by applying the scaled 5–95% uncertainty range from Thornhill et al. (2021b) to this best-estimate value. This means that some precursor ranges are outside the range of that described by Thornhill et al. (2021b), though only seven models (fewer for some precursors) provided the necessary experiments in Thornhill et al. (2021b), and thus AerChemMIP represents a small ensemble of opportunity.

| Precursor | Contribution to ozone ERF |
|---:|:---|
| CH$_4$ | $2.35 \times 10^{-4} \pm 6.18 \times 10^{-5}$ W m$^{-2}$ ppb$^{-1}$ |
| N$_2$O | $1.18 \times 10^{-3} \pm 4.73 \times 10^{-4}$ W m$^{-2}$ ppb$^{-1}$ |
| Chlorinated and brominated GHGs | $-5.48 \times 10^{-5} \pm 1.20 \times 10^{-4}$ W m$^{-2}$ (ppt CFC-11 EESC)$^{-1}$ |
| CO | $2.34 \times 10^{-5} \pm 1.33 \times 10^{-4}$ W m$^{-2}$ (MtCO yr$^{-1}$)$^{-1}$ |
| VOCs | $2.73 \times 10^{-4} \pm 3.67 \times 10^{-4}$ W m$^{-2}$ (MtVOC yr$^{-1}$)$^{-1}$ |
| NOx | $1.19 \times 10^{-3} \pm 1.17 \times 10^{-3}$ W m$^{-2}$ (MtNO$_2$ yr$^{-1}$)$^{-1}$ |

**Table 3.** Distributions of the contributions to the ozone ERF sampled in `fair-calibrate` v1.4.1. Uncertainty ranges are shown as 90% ranges and sampled from a Gaussian.





### 3.2.6 ERF scalings

Forcing uncertainties in ERFari, ERFaci and ozone are sampled from the contribution to total forcing from their precursor species as described in previous sections. For other major categories of forcings, we use the IPCC AR6 ranges (Forster et al., 2021) as relative uncertainty factors to scale the ERF (table 4).

    For $CO_2$, we use the sampled $F_{4\times CO_2}$ value from the climate response calibration and perform a quantile mapping to derive a scaling factor for $CO_2$ forcing that is Gaussian. While this does not preserve the shape of the $F_{4\times CO_2}$ distribution kernel, it

does map low $4 \times CO_2$ forcings to low $CO_2$ scalings and vice versa.

| Forcing | Relative uncertainty and distribution |
|---:|:---|
| $CO_2$ | ±0.12, Gaussian |
| $CH_4$ | ±0.20, Gaussian |
| $N_2O$ | ±0.14, Gaussian |
| Halogenated GHGs | ±0.19, Gaussian |
| Stratospheric water vapour from methane oxidation | ±1.00, Gaussian |
| Land use change | ±0.50, Gaussian |
| Volcanic | ±0.25, Gaussian |
| Solar amplitude | ±0.50, Gaussian |
| Solar linear trend 1750–2019 | +0.01 (−0.06 to +0.08) W m$^{-2}$, Gaussian |
| BC on snow | 5th and 95th percentiles at (0.00, 2.25), skew-normal |
| Contrails* | 5th and 95th percentiles at (0.33, 1.72), skew-normal |

**Table 4.** Forcing scaling factors used to translate the raw best estimate from FaIR to IPCC assessed uncertainty ranges (Forster et al., 2021). Scaling uncertainty ranges are 5–95%. Except for solar trend, median distribution values are 1. *Contrails forcing is not used in v1.4.0 and v1.4.1 but is included in other versions.

## 3.3 Constraining

The 1.6 million member prior ensemble of FaIR climate projections is compared to historical observations and assessments of climate metrics from either the IPCC AR6 (Forster et al., 2021) or their updates based on more recent data (Forster et al., 2023).

### 3.3.1 Step 1: Root-mean-squared difference with respect to historical

The root-mean-square (RMS) difference of each ensemble member's GMST anomaly projection compared to historical for 1850–2022 is used as a simple pass/fail criterion for ruling out parameter sets that are inconsistent with historical observed warming. Ensemble members that have an RMS difference of greater than 0.17°C are rejected. The mean of four GMST





datasets (HadCRUT5, Berkeley Earth, NOAAGlobalTemp and Kadow) from Forster et al. (2023) is used as the historical
GMST dataset for comparison. 0.17°C is a somewhat arbitrary choice, which balances sufficient variability in the historical
record to allow for observational uncertainty with the need for projections that are true to observations. By design, this threshold
roughly reproduces the uncertainty range in present-day GMST relative to pre-industrial assessed by the IPCC (Gulev et al.,
2021), whereas a more stringent threshold may over-constrain both the historical observational uncertainty and scope for
future climate projection uncertainty (fig. 6). Internal variability is switched on for this historical comparison, to allow for
the possibility that the historical record can be well-simulated by chance in mean-state climate configurations that would be
warmer or cooler than expected (e.g. a strong pattern effect; Andrews et al. (2018)). This step reduces the ensemble size from
1.6 million to 224,342, ruling out around 86% of the original ensemble.

Figure 6 compares the ten ensemble members with the lowest RMSE relative to observations (blue; RMSE ≈ 0.10°C) with
the ten largest RMSE members that still meet the RMSE constraint (red; RMSE ≈ 0.17°C). Figure 6 shows that runs with low
internal variability tend to result in the closest correspondence with historical observed temperature, and therefore the final
ensemble could be biased towards ensemble members with smaller variability. A formal analysis of the internal variability
characteristics in relation to observations is not performed in this version of `fair-calibrate`, though could be added to
the constraining criteria in the future.

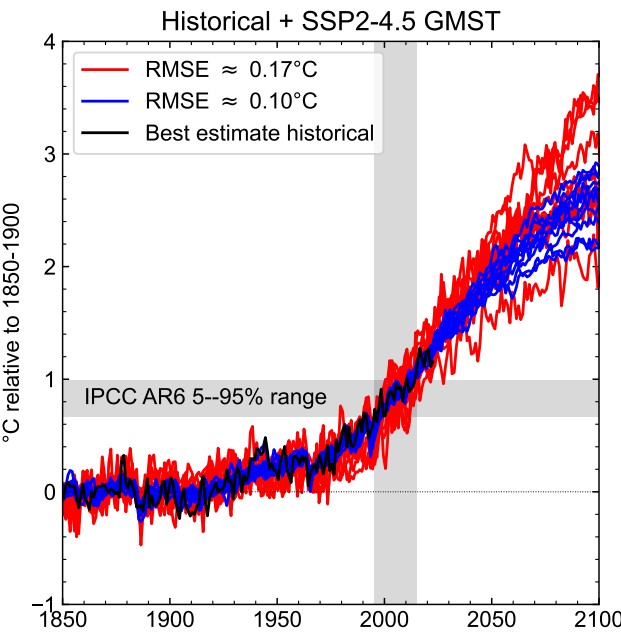

**Figure 6.** Comparison of the 10 ensemble members with the smallest RMSE error (blue) compared to the historical best estimate GMST
from Indicators of Global Climate Change 2022 (Forster et al., 2023) (black) with the 10 with the largest RMSE (red) that passed this first
historical constraining step.



### 3.3.2 Step 2: Reweighting based on observed and assessed climate metrics

The second constraining step takes the ensemble members that passed the RMSE threshold and simultaneously fits the projections to eight target distributions (fig. 7). For each target distribution, either a Gaussian (if symmetric) or skew-normal (if asymmetric) continuous probability distribution is constructed from the 5th, 50th and 95th percentiles of the variable's uncertainty range. As a three-parameter distribution, a skew-normal can uniquely fit three specified quantiles. The percentiles of the target distributions are shown in the first eight rows of table 5. Emergent parameters (ECS, TCR, and aerosol forcing ranges) are taken from the IPCC AR6 WG1 Chapter 7 (Forster et al., 2021), and updated climate observations (GMST, OHC and $CO_2$ concentrations) are taken from the Indicators of Global Climate Change 2022 (Forster et al., 2023).

The ensemble size in the final reweighted constrained distribution is a user choice. Typically ensemble sizes of a few hundred to a few thousand are used for projections using reduced-complexity models (Nicholls et al., 2021), which allows for full exploration of the uncertainty space while keeping the number of simulations small enough to allow for efficient computation. For the final posterior distribution in calibrations v1.4.0 and v1.4.1 we select 841 ensemble members. 841 is one more than a highly composite number and allows many quantiles of the full distribution to correspond to a single ensemble member at each point in time. The 841 ensemble members are drawn from an effective ensemble size of 4356. To enable the target distributions to be well-fit, we find that in practice an effective ensemble size that is at least five times larger than the desired final posterior ensemble is ideal.

The evolution of GMST projections from the prior ensemble, to the historical RMSE constraint, and finally the reweighted constrained ensemble is shown in fig. 8. The prior ensemble allows for a wide range of projections, the majority of which are clearly incompatible with historical GMST (fig. 8a). The RMSE threshold step, alongside producing historically reasonable projections, substantially reduces the range in projected future warming (fig. 8b). However, low and particularly very high future warmings pass the historical RMSE constraint. The reweighting step provides a narrower band on historical warming as well as reducing the spread in future warming further (fig. 8c). The 5–95% ranges of future warming are similar between the RMSE constraint and the reweighted posterior, but the latter distribution constrains out much of the warm and cool tails of the distribution that passes the RMSE constraint.

Figure 9 shows the distributions of the 45 parameters used to construct the prior samples (green histograms) and the reweighted posterior (red histograms). Table S3 lists the parameters and the part of the model that is being affected as well as its location within the paper. For some distributions, the constraining steps create posteriors that are differently shaped to the priors. Sometimes this is by design. For example, $\kappa_1$, the climate feedback parameter, is inversely related to ECS, and the IPCC constraint downweights the likelihood of "hot" combinations (noting that the prior distribution is constructed from CMIP6 models, many of which have higher climate sensitivity than the 95th percentile of 5°C assessed in IPCC AR6). Occasionally, distributions are multi-modal such as the parameters that define the ERFaci shape, due to the model calibrations themselves spanning several orders of magnitude.





# 4    Characteristics of calibrations v1.4.1 and v1.4.0

As a demonstrative case we show GMST projections for the eight Tier 1 and Tier 2 SSPs using the harmonized emissions scenarios in fig. 10 using calibration v1.4.1. Alongside SSP projections, we use the posterior parameter sets and run concentration-driven runs with a compound 1% per year $CO_2$ concentration increase for 140 years. This allows determination of the airborne

fraction of $CO_2$ at the time of doubling (70 years) and quadrupling (140 years), an estimate of the TCRE obtained at the point of crossing 1000 GtC of emissions, and a CMIP-consistent approach to calculating TCR (fig. S1).

For the emissions-driven SSP scenarios, the large-scale warming behaviour is in line with expectations, with high emissions scenarios such as SSP5-8.5 and SSP3-7.0 showing several degrees of warming over the next two centuries, and lower emissions scenarios warming less. Scenarios where $CO_2$ emissions turn net negative (SSP1-1.9, SSP1-2.6 and SSP5-3.4-overshoot) show

peak and decline behaviour in the ensemble median, though some extreme high ensemble members continue to warm beyond net zero owing to a positive zero emissions commitment (Palazzo Corner et al., 2023).

For a more rigorous comparison we compare the reweighted constrained posterior from `fair-calibrate` v1.4.1 to the assessed ranges in the AR6 WG1 assessments in table 5 (see Cross Chapter Box 7.1 in Forster et al. (2021) and Smith et al. (2021a)). The first eight rows of the table are the distributions used to reweight the posterior. By design, the fit to the target

distribution in these eight cases is very good (in most cases, no shading in the Difference columns in table 5). The slight disagreement with the lower bound of the transient climate response is due to the IPCC assessment of the lower end of the *very likely* range of TCR being lower than the lowest TCR in any of the CMIP6 models which are used to create the prior distribution sample. A better fit to the IPCC assessed range could be achieved by increasing the samples in the prior TCR distribution at the lower end. The disagreement in the upper bound of ERFaci is large in absolute terms but small in relative

terms. Similarly, no comparison for the upper bound of ERFari is provided to avoid division by zero.

The remaining assessed ranges in table 5 are used for validation and sense-checking. FaIR under-predicts and provides a narrower range of airborne fraction at $2\times CO_2$ and $4\times CO_2$, and TCRE. However, the sensitivities of the carbon cycle feedbacks in FaIR are already well-constrained by comparison of the 1750 to 2022 $CO_2$ emissions with observed concentrations, which places a tight bound on the historical cumulative airborne fraction. The IPCC assessment of airborne fraction is taken from

CMIP6 idealised 1pctCO2 runs and is entirely CMIP6-model-based (Arora et al., 2020), and emissions-driven CMIP6 ESMs do not reproduce present day $CO_2$ concentrations as tightly as our observational constraint (Lee et al., 2021). In idealised frameworks, TCRE is proportional to the product of airborne fraction and TCR (Jones and Friedlingstein, 2020). The IPCC TCRE assessment is wider than the product of the TCR and airborne fraction individual assessments in quadrature and as such, distribution fitting to the AR6 assessed ranges of TCR, airborne fraction and TCRE simultaneously is not possible.

We also compare the emissions-driven SSP temperature projections in FaIR to the assessed ranges from the IPCC AR6 WG1 (Lee et al., 2021). For the strong mitigation scenarios SSP1-1.9 and SSP1-2.6, the SSP warming is above the IPCC assessed ranges, particularly at the 95th percentile. We suggest three reasons. Firstly, concentration (not emissions) driven runs were used to derive the IPCC warming ranges, which excludes the impact of carbon cycle sensitivity uncertainty on a future spread in $CO_2$ concentrations and thus over-constraining the uncertainty range. Secondly, the spread in aerosol forcing in our




calibration is larger than in CMIP6 (Smith et al., 2020) and the constrained emulator used in the IPCC (Forster et al., 2021). Thirdly and most importantly, the starting point for the future scenario is now 2023 rather than 2015, and emissions have been higher in reality over the last eight years than in the original SSP1-1.9 and SSP1-2.6 scenarios. The influences of the first and third effects can be visualised by comparing the emissions and projected concentrations of $CO_2$ between v1.4.0 and v1.4.1 (fig. 11). Figure 11a also confirms that $CO_2$ emissions in the recent past can be well-approximated with the SSP2-4.5 scenario.

Conversely, the high emissions SSP3-7.0 and SSP5-8.5 scenarios are projected to warm less in `fair-calibrate` v1.4.1 compared to the assessments in AR6 WG1. As for the low emissions scenarios, the high emissions scenarios have started to diverge from recent history for $CO_2$ (fig. 11a). The emissions-driven projections from FaIR tend to result in lower $CO_2$ concentrations than in the equivalent CMIP6 scenarios (derived using MAGICC6), likely due to the carbon cycle sensitivities being higher in the CMIP6 calibration of MAGICC6 (fig. 11b).

We show the comparison to the AR6 assessed ranges for `fair-calibrate` v1.4.0 in table S4. In general, these are closer to the IPCC assessments than for v1.4.1, particularly for SSP warming projections, noting that the SSP emissions start in 2015. One reason for the "narrowing" of projections in v1.4.1 (lower scenarios are warmer, higher scenarios are cooler) is the additional eight years of near constant $CO_2$ emissions for the 2015–2022 period in the harmonized scenarios used, reducing the range of climate outcomes in 2100 that are possible with SSP scenarios that satisfy recent historical constraints. One important 495 corollary of this is that median peak warming in the updated, harmonized SSP1-1.9 scenario is 1.69°C in calibration v1.4.1 compared to 1.57°C in v1.4.0, meaning that is is now very unlikely that any realistic mitigation scenario could limit warming to 1.5°C with no or low overshoot (Dvorak et al., 2022).

## 5 Conclusions

This paper describes a package, `fair-calibrate`, that calibrates the responses of the FaIR simple climate model to com-500 plex Earth System models, generates a large Monte Carlo ensemble sample, and constrains the results to observations and expert assessments. We claim that a rigorous calibration process that produces ensemble results that are consistent with historically observed climate is a necessary (though not sufficient) condition for trustworthy future climate projections using simple climate models.

We demonstrate two calibrations in this paper: v1.4.1 based on the most up-to-date estimates of all emitted greenhouse gases 505 and short-lived climate forcers, and v1.4.0 which uses emissions time series prepared for CMIP6 and AR6 (but are now becoming increasingly outdated). The two different versions presented in this paper produce notably different future projections. The choice of calibration to use depends on user application, and care should be taken to ensure the correct calibration is used for the supplied emissions. Additional calibrations using alternative emissions time series and/or constraints can be generated under similar procedures to that described in the paper and accompanying code. Furthermore, the calibration mechanism could 510 be extended to account for different constraints, for example on TCRE, the zero emissions commitment, warming rates, or future scenario warming. Addition of further constraints should be done with care to ensure internal consistency, particularly when correlated with other constraints, and would likely require a larger prior ensemble size or alternative sampling strategy.



We intend to produce operational updates to `fair-calibrate` on at least an annual basis. A calibration could be updated based on new climate constraints such as the anticipated yearly updates to Indicators of Global Climate Change (Forster et al.,
2023), new source emissions (such as an expected update to CEDS, which will update SLCF emissions to 2022), or new future emissions scenarios (such as those from Network for Greening the Financial System). Operationally updated calibrations of emulators and scenarios that reflect the latest scientific knowledge, from which near-future warming can be assessed, will be a beneficial tool in tracking progress towards Paris Agreement aims.

*Code and data availability.* Code is available at https://github.com/chrisroadmap/fair-calibrate and is archived, along with intermediate and
output data, at https://doi.org/10.5281/zenodo.10566813 (Smith, 2024).

*Author contributions.* CS led development of the `fair-calibrate` package and led the paper writing. DC developed the stochastic three-layer energy balance model that is the default climate response module in FaIR v2.1, and the `EBM R` package that calibrates it. HBF provided processed annual global mean data from CMIP6 models used in the calibration step. ZN and MM wrote the Bayesian weighting code. NL, SJ, CS and CM developed the FaIR model from v2.0 onwards with support from MA. AIP helped to rectify an inconsistency in
the definitions of TCR and TCRE in an earlier calibration version.

*Competing interests.* The authors declare no competing interests.

*Acknowledgements.* CS acknowledges funding from a NERC/IIASA Collaborative Research Fellowship (NE/T009381/1) and the European Commission under grant agreement no. 101081661 (WorldTrans).



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





**Figure 7.** Comparison of distributions of key climate metrics (table 5) in each step of the constraining process. The prior distributions from the 1.6 million member prior ensemble are in green. The first constraining step using the RMSE comparison to historical temperature is in purple. The second constraining step that reweights each distribution to its target is in red. The target distribution is in black. The goal is for the red distribution to be as close as possible to the black across all metrics.



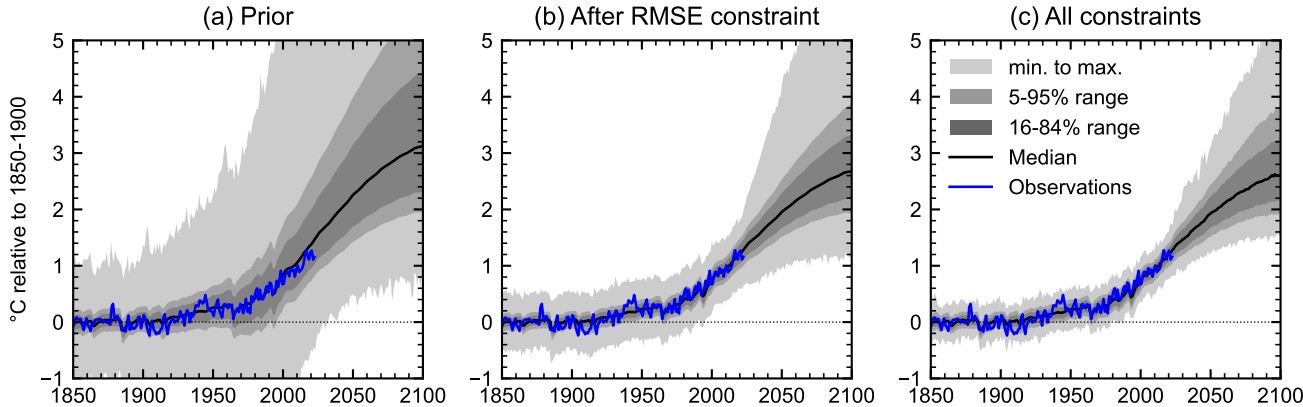

**Figure 8.** Progression of projections using the historical + harmonized SSP2-4.5 emissions for (a) all prior ensemble members, (b) the RMSE < 0.17°C first constraining step and (c) the final reweighted and constrained posterior. In each plot, progressively darker shaded regions correspond to the minimum–maximum, 5–95%, 16–84% ranges, black line is ensemble median and blue line is historical best estimate GMST from Indicators of Global Climate Change 2022 (Forster et al., 2023)







**Figure 9.** Prior (green) and reweighted posterior (red) distributions of the 45 parameters sampled. For a description of what the parameters correspond to, refer to table S3.



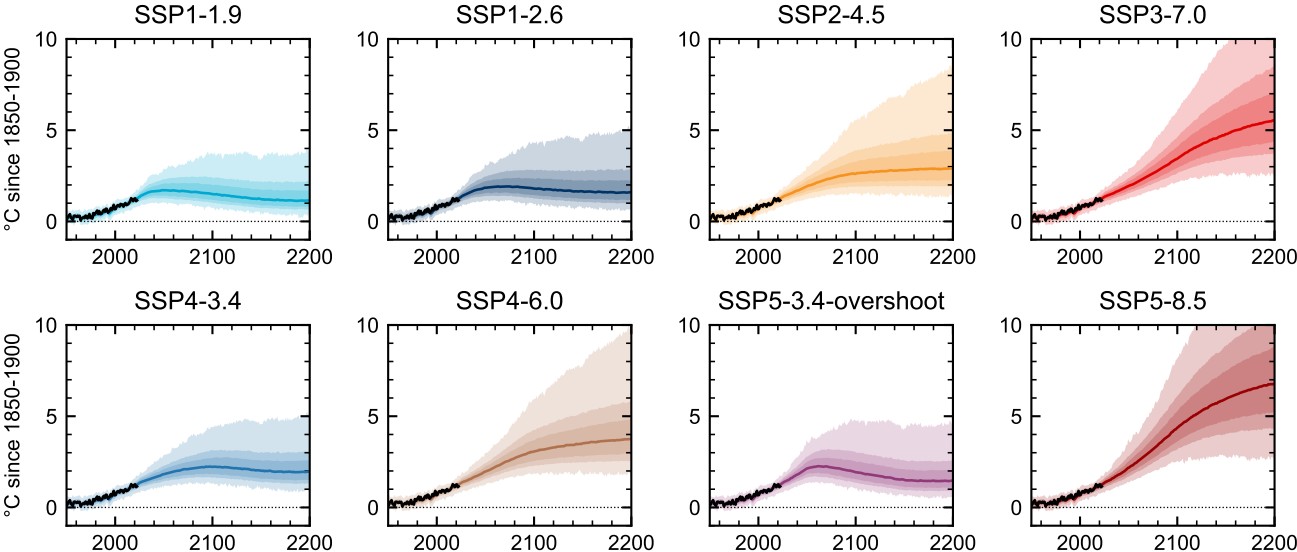

**Figure 10.** Projections using the weighted posterior for the eight main SSPs scenarios. Shaded ranges are (from dark to light) minimum to maximum, 5–95% and 16-84% of the distribution. Solid lines are distribution medians, black lines are best estimate historical warming.



| Metric | Target 5% | Target 50% | Target 95% | Reweighted posterior 5% | Reweighted posterior 50% | Reweighted posterior 95% | Relative difference 5% | Relative difference 50% | Relative difference 95% | Fit? |
|---|---|---|---|---|---|---|---|---|---|---|
| ECS (K) | 2.00 | 3.00 | 5.00 | 2.01 | 2.96 | 4.99 | +1% | −1% | 0% | Yes |
| TCR (K) | 1.20 | 1.80 | 2.40 | 1.31 | 1.79 | 2.38 | +9% | 0% | −1% | Yes |
| GMST 2003–2022 rel. 1850–1900 (K) | 0.87 | 1.03 | 1.13 | 0.86 | 1.03 | 1.13 | −1% | 0% | 0% | Yes |
| EEU 2020 rel. 1971 (ZJ) | 356.8 | 465.3 | 573.8 | 355.5 | 466.9 | 587.3 | 0% | 0% | +2% | Yes |
| Aerosol ERF 2005–2014 rel. 1750 (W m$^{-2}$) | −2.0 | −1.3 | −0.6 | −1.94 | −1.27 | −0.56 | −3% | −2% | −7% | Yes |
| ERFari 2005–2014 rel. 1750 (W m$^{-2}$) | −0.6 | −0.3 | 0.0 | −0.58 | −0.30 | 0.00 | −3% | −2% | | Yes |
| ERFaci 2005–2014 rel. 1750 (W m$^{-2}$) | −1.7 | −1.0 | −0.3 | −1.66 | −0.96 | −0.35 | −2% | −4% | +15% | Yes |
| $CO_2$ concentration 2022 (ppm) | 416.2 | 417.0 | 417.8 | 416.1 | 417.0 | 417.8 | 0% | 0% | 0% | Yes |
| WMGHG ERF 2019 rel. 1750 (W m$^{-2}$) | 3.03 | 3.32 | 3.61 | 3.01 | 3.32 | 3.62 | −1% | 0% | 0% | |
| $CH_4$ ERF 2019 rel. 1750 (W m$^{-2}$) | 0.43 | 0.54 | 0.65 | 0.45 | 0.56 | 0.66 | +4% | +3% | +1% | |
| Airborne fraction at 2×$CO_2$* | 0.43 | 0.53 | 0.63 | 0.47 | 0.48 | 0.49 | +10% | −9% | −22% | |
| Airborne fraction at 4×$CO_2$* | 0.44 | 0.60 | 0.76 | 0.47 | 0.55 | 0.59 | +7% | −8% | −22% | |
| TCRE* (K (1000 GtC)$^{-1}$) | 0.58 | 1.65 | 2.72 | 1.09 | 1.47 | 1.92 | +88% | −11% | −29% | |
| SSP1-1.9 2021–2040 rel. 1995–2014 (K) | 0.38 | 0.61 | 0.85 | 0.38 | 0.65 | 0.97 | +1% | +7% | +14% | |
| SSP1-1.9 2041–2060 rel. 1995–2014 (K) | 0.40 | 0.71 | 1.07 | 0.44 | 0.83 | 1.39 | +9% | +17% | +30% | |
| SSP1-1.9 2081–2100 rel. 1995–2014 (K) | 0.24 | 0.56 | 0.96 | 0.24 | 0.73 | 1.48 | 0% | +31% | +54% | |
| SSP1-2.6 2021–2040 rel. 1995–2014 (K) | 0.41 | 0.63 | 0.89 | 0.40 | 0.67 | 0.97 | −1% | +6% | +9% | |
| SSP1-2.6 2041–2060 rel. 1995–2014 (K) | 0.54 | 0.88 | 1.32 | 0.57 | 0.99 | 1.55 | +5% | +12% | +17% | |
| SSP1-2.6 2081–2100 rel. 1995–2014 (K) | 0.51 | 0.90 | 1.48 | 0.47 | 1.02 | 1.81 | −7% | +13% | +22% | |
| SSP2-4.5 2021–2040 rel. 1995–2014 (K) | 0.44 | 0.66 | 0.90 | 0.41 | 0.65 | 0.91 | −6% | −2% | +1% | |
| SSP2-4.5 2041–2060 rel. 1995–2014 (K) | 0.78 | 1.12 | 1.57 | 0.72 | 1.09 | 1.57 | −7% | −3% | 0% | |
| SSP2-4.5 2081–2100 rel. 1995–2014 (K) | 1.24 | 1.81 | 2.59 | 1.06 | 1.71 | 2.66 | −14% | −6% | +3% | |
| SSP3-7.0 2021–2040 rel. 1995–2014 (K) | 0.45 | 0.67 | 0.92 | 0.41 | 0.64 | 0.89 | −8% | −5% | −3% | |
| SSP3-7.0 2041–2060 rel. 1995–2014 (K) | 0.92 | 1.28 | 1.75 | 0.79 | 1.12 | 1.54 | −15% | −13% | −12% | |
| SSP3-7.0 2081–2100 rel. 1995–2014 (K) | 2.00 | 2.76 | 3.75 | 1.63 | 2.31 | 3.18 | −19% | −16% | −15% | |
| SSP5-8.5 2021–2040 rel. 1995–2014 (K) | 0.51 | 0.76 | 1.04 | 0.45 | 0.69 | 0.98 | −11% | −9% | −5% | |
| SSP5-8.5 2041–2060 rel. 1995–2014 (K) | 1.08 | 1.54 | 2.08 | 0.94 | 1.37 | 1.97 | −11% | −9% | −5% | |
| SSP5-8.5 2081–2100 rel. 1995–2014 (K) | 2.44 | 3.50 | 4.82 | 2.12 | 3.09 | 4.37 | −13% | −12% | −9% | |

**Table 5.** Comparison of IPCC AR6 WG1 (Forster et al., 2021; Lee et al., 2021; Gulev et al., 2021) or updated (Forster et al., 2023) observational and assessed distributions ("Target" columns), the distributions of the posterior from calibration v1.4.1 ("Reweighted posterior"), and the relative percentage difference. Distributions denoted with * were assessed as *likely* ranges in IPCC AR6 WG1, interpreted as ± 1 s.d., and have been converted to 5–95% ranges here for consistency with other values. Metrics with "Yes" in the Fit column are part of the multiple constraining described in section 3.3.2. Dark shading is more than 20% from the target for upper and lower ranges and 10% for the central. Paler shading is more than 10% from the target for upper and lower ranges and 5% for the central.

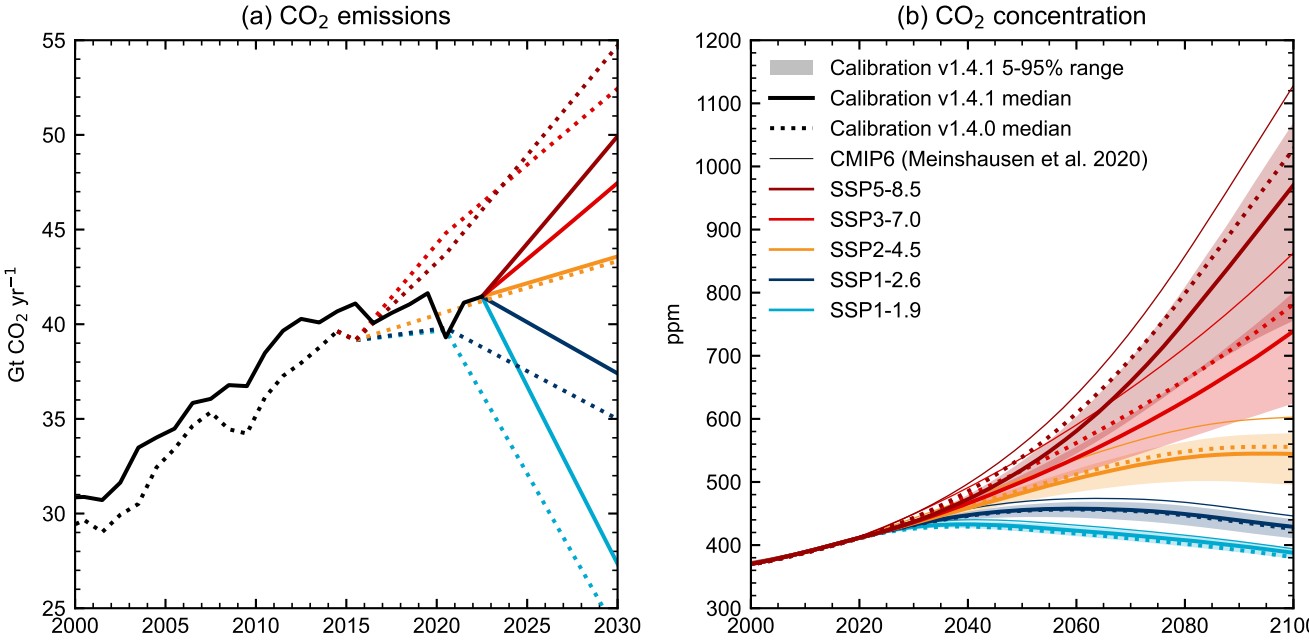

**Figure 11.** (a) $CO_2$ emissions in calibration v1.4.1 (GCP 2023 v1.0 up to 2022, harmonized SSP projections after) in solid lines, calibration v1.4.0 (RCMIP v5.1.0) in dotted lines. (b) Median $CO_2$ concentration projections from v1.4.1, v1.4.0, and CMIP6 (thin lines). 5–95% range from v1.4.1 shown in shaded regions.