# Peer review of "fair-calibrate v1.4.1: calibration, constraining and validation of the FaIR simple climate model for reliable future climate projections"

_EGUsphere, 2024_

## Author Comment (AC1)

**Reviewer #1**

Paper provides rather detail descriptions of used climate model and parameter calibration procedure.

However, in my view it requires some clarifications and revision before publication.

Thank you for your time reviewing the manuscript. We describe the clarifying revisions that we will make below and trust that the improvements we make will be satisfactory.

Authors say in Abstract:

"We show that two very different future projections to a given emission scenario can be obtained using emissions from the IPCC Sixth Assessment Report (AR6) (fair-calibrate v1.4.0) and from updated emissions datasets through 2022 (fair-calibrate v1.4.1) for similar climate constraints in both cases."

However, simulations presented in the paper were done not only using different parameters distributions but also very different emission scenarios (Fig 11). Moreover, in section 4 authors mentioned difference in future emission while explaining difference between results of v1.4.1 and v1.4.0 ensembles.

I think simulations with v1.4.0 set of model parameters need to be redone with different emissions scenarios. As I understand, historical simulations used to derive final parameters distributions were run until 2022 using historical plus SSP2-4.5 from AR6. It looks logical to produce another set of harmonized SSP emissions using 2022 SSP2-4.5 emissions instead of historical and repeat v1.4.0 ensemble with these emissions. It should make comparison cleaner. It also would be interesting to mention distribution of which model parameters are most sensitive to changing historical emissions.

If we understand correctly, you suggest running the same emissions scenario (e.g. the IPCC AR6 emissions) in the two different calibration versions to explore differences that would be attributable to changes in emissions and changes in calibration. This is a good idea. These contributions are something that we plan to explore in a new international collaboration led by IIASA, which will use annual updates of the calibrations of FaIR and MAGICC. We propose that this is beyond the scope of the current study. The reason is that the emissions are themselves a vital part of the calibration set. For all greenhouse gas species, the observed atmospheric concentrations (which are measurable and known to have low uncertainty) are calibrated to be reproduced as closely as possible from precursor emissions. The emissions are the datasets that are more uncertain. We demonstrate this throughout the paper. For $CO_2$, we require that present day concentrations are correctly reproduced. The cumulative historical emissions in the latest iteration of Global Carbon Budget, used in v1.4.1, are about 5% higher than the cumulative emissions used in AR6 (v1.4.0) (see fig. S4). Therefore, the carbon cycle feedback strengths are adjusted to ensure that the correct concentrations are reproduced. If we used Global Carbon Project emissions with the v1.4.0 calibration, we

would be overestimating the present-day $CO_2$ concentrations, and the fitness for purpose of the model calibration for future projections could be, rightly, questioned.

Specific comments.

Page 7. Line 185. Reference to fig S1 is confusing.

Confusing and indeed incorrect. Later in the paragraph we mention the parameter values listed in table S1, which is the intention. We will drop the reference to fig. S1, thanks.

Page 10 line 254.

concentrations of methane are in approximate equilibrium with pre-industrial concentrations.

It should be:

concentrations of methane are in approximate equilibrium with pre-industrial emissions.

Correct, thank you for spotting this. Will be updated in revision.

line 267 should be fig. S3,

Correct, will be updated in revision. On re-reading the sentence, we should also replace "lifetimes, historical and future calibrations" with "lifetimes, historical and future concentrations".

line 269 should be fig S4b

Correct, will be updated

Lines 271-273

"As 1750 emissions are subtracted from the total to report changes away from a pre-industrial equilibrium, the change in emissions (1750–2022) in v1.4.1 from PRIMAP-Hist is smaller than in v1.4.0, leading to longer atmospheric lifetimes necessary to reproduce concentrations."

According to Fig. S4b, 1750 CH4 emissions are larger in v1.4.1 than in v1.4.0. If 1750 CH4 concentration are identical in both cases, it will require large CH4 lifetime is for v1.4.1 for concentration to be in equilibrium with emissions. Please comment.

This raises an interesting point. In v1.4.0 (with AR6 emissions), $CH_4$ emissions in 1750 are around 19 $TgCH_4$/yr. In v1.4.1 (with PRIMAP-Hist replacing the RCMIP database for fossil and agricultural emissions), total $CH_4$ emissions in 1750 are about 38 $TgCH_4$/yr. Therefore, in order to maintain equilibrium concentrations in 1750, the atmospheric

lifetime would need to be shorter in v1.4.1 than in v1.4.0. Our calibrations show the opposite, with 1750 lifetimes around 16.8 years in v1.4.1 and 10.0 years in v1.4.0. This is because our lifetimes are calculated on the change in methane emissions over the historical period and fit to this, rather than starting from an equilibrium value in 1750.

There are two limitations to the methane treatment in FaIR at the moment that leads to this. The first is that, starting in FaIR v2.0.0, methane concentrations are calculated as a perturbation from pre-industrial due to anthropogenic emissions only, rather than accounting for all (anthropogenic plus natural) sources of methane. Natural sources are significant, and include wetlands, termites and permafrost, and are associated with a climate feedback. It would be more appropriate to account for all sources and sinks of methane in the atmosphere and model the full atmosphere lifetime. This is an area of development for FaIR. The second limitation is that the 1750 methane concentration is not an equilibrium value, and there has been a steady increase in concentrations over the Holocene, which is likely due to agricultural activities. Looking at the CMIP6 data that goes back to 1 CE (supplementary data to Meinshausen et al., 2017, at https://view.officeapps.live.com/op/view.aspx?src=https%3A%2F%2Fwww.climatecolleg e.unimelb.edu.au%2Ffiles%2Fsite1%2Fdocs%2F11%2FSupplementary_Table_UoM_G HGConcentrations-1-1-0_annualmeans_v2.xls&wdOrigin=BROWSELINK), concentrations increase from ~650 ppb in the first decade of the millennium to ~730 ppb around 1750. It's also likely that the agricultural revolution and measurable increases in concentrations started a long time before this (and again, there is the question of whether natural fluxes have varied across this period). We would ideally therefore want emissions going back to 1 CE in order to calibrate to this period, but even this is imperfect due to the likelihood that concentrations were not in equilibrium even then.

Ultimately this will be a function of the calibration and constraining method and the target outcome of the user. For contemporary climate change and future projection problems, which is FaIR's primary use case, it is a necessary condition to get the historical climate correct (here "historical" refers to post-1750), and hence we prioritise this. Should anybody want to use FaIR to investigate different periods of Earth's history, it may require a different calibration, to focus on conditions that are more suitable to answer the desired questions.

In summary, in the revised paper we will highlight this apparent discrepancy and use it as further justification for improving the methane treatment in future model development.

Page 20.

"841 is one more than a highly composite number and allows many quantiles of the full distribution to correspond to a single ensemble member at each point in time."

Needs explanation.

We will add a bit more explanation here in the revision. 841 is not a fixed number and is a user choice. Older calibration versions used 1001 posterior members. Simple climate models generally range from a few hundred to a few thousand ensemble members

(Nicholls et al., 2021). The "requirement" that the posterior size is one larger than a highly composite number is merely a preference of the authors. The main important features of the posterior ensemble are that (1) it should be large enough to give a smooth coverage of the distributions of each constraining parameter and fully sample uncertainty; and (2) small enough that it can be generated from an unbiased sample of candidate ensemble members after likelihood weighting. Condition (1) generally imposes a lower bound of around 500 but is not a hard-and-fast rule. Condition (2) is mentioned in the paper, in that the desired posterior should be no more than about 20% as large as the likelihood-weighted sample (this we found by trial and error). If (1) and (2) cannot be simultaneously met, a bigger prior, differently sampled prior, or relaxation of one or more constraints is necessary.

Page 21. Line 464

"The disagreement in the upper bound of ERFaci is large in absolute terms but small in relative terms."

It seems to be other way around.

Yes, you are correct; excellent spot. We will fix this in the revision.

Page 21. Line 477

"Firstly, concentration (not emissions) driven runs were used to derive the IPCC warming ranges, which excludes the impact of carbon cycle sensitivity uncertainty on a future spread in CO2 concentrations and thus over-constraining the uncertainty range."

IPCC ranges for SSP scenarios, shown in tables 5 and S4, are not from AR6 Table 4.2, which shows ranges based on CMIP6 model simulations, but from Table 4.5, which shows Assessment results for 20-year averaged GSAT change, based on multiple lines of evidence.

Correct: our intention is to use results from table 7.SM.4 and Cross Chapter Box 7.1, which are the same for global mean surface temperature projections that are reported in IPCC AR6 Chapter 4 Table 4.5, and are from multiple lines of evidence besides CMIP6 models. Additionally, table 7.SM.4 & tables 5 and S4 in the paper contain additional constraints from various chapters of the IPCC WG1 report (or their updates in Forster et al. 2023). In AR6, this is the approach that we (Smith & Nicholls) used to constrain the simple climate models. Therefore, we believe that our intent was correct by referring to them as IPCC warming ranges rather than CMIP6 ranges. However, we appreciate the confusion and will be more specific in the revision.

**Reviewer #2**

The study proposes a new version of the FaIR simple climate model. It uses a Bayesian framework to estimate the parameters of the FaIR model and clearly explains the differences between fair-calibrate v1.4.0 and fair-calibrate v1.4.1. I enjoyed studying the entire manuscript. I have a few comments listed below.

Thank you for the positive comments. We trust that the revisions that we implement following your comments below will lead to an improved manuscript.

Can you please provide a flowchart of the working principle of fair-calibrate v1.4.1?

A good idea. We will do this.

If I understood correctly, the authors have used kernel density estimation as a prior distribution in the Bayesian framework. KDE can be very sensitive to outliers and data sparsity. Could you please discuss this in the manuscript?

That's correct, we used kernel density estimates. Data sparsity is a real issue; the climate response (49 models) is reasonably well-populated but the aerosol forcing and carbon cycle tunings (13 and 11 models) suffer from sparsity, and the aerosol parameters span several orders of magnitude. The models that we have to calibrate to are an ensemble of opportunity, so do not sample all possible realistic model responses to forcings.

We will discuss the limitations of KDE in the manuscript. We believe it is better than alternatives, as most smooth analytic distributions are difficult to generalise to various shapes (particularly left skew and multi-modality). An alternative would be to not draw distributions at all and select one model parameter set at random for each ensemble member, however this suffers from the ensemble of opportunity limitation.

Can you please explain how you preserve the correlation structure between parameters while sampling parameters (section 3.2)?

This results from the SciPy library implementation of a Gaussian kernel density estimate (scipy.stats.gaussian_kde), for which the documentation page is at https://docs.scipy.org/doc/scipy/reference/generated/scipy.stats.gaussian_kde.html. The worked example on this page (scipy v1.14.0, last accessed 30 June 2024) shows the preservation of correlation structure between samples drawn from two random variables. In out paper this is generalised to higher dimensional spaces, for example 11 dimensions in drawing the parameters that drive the stochastic three layer energy balance model.

A very similar method is demonstrated for sampling aerosol forcing parameters, showing the pairwise correlation between parameter sets, in fig. 4 of Smith et al. 2021, https://agupubs.onlinelibrary.wiley.com/doi/10.1029/2020JD033622.

The mathematical details are in the Scott (1992) textbook reference, which we do not reproduce since this is most likely too much technical detail for the intended audience.

In the revision however, it could be worth us referencing the SciPy documentation, such that readers know precisely what protocol was followed.

Can you please provide a table indicating the prior distribution of different parameters and the parameters of prior distributions?
Yes, we can do that.

Can you please comment on the sensitivity of the parameters sampled using the MCMC approach?
In this paper we do not use an MCMC approach, though we understand why the reviewer may think that we did.

Is there any reason why you selected gaussian distribution (section 3.3.2)? Have you performed any goodness of fit tests?

We'll provide a bit more commentary on this in the revision.

In section 3.3.2 we are looking to fit the output from the first step to constraint distributions for the final posterior that (in the IPCC assessment that was the motivation for this work) were defined in terms of their 5th, 50th and 95th percentiles, but the actual distributions that we should fit to were not prescribed. We are looking for distributions that are uniquely defined from three degrees of freedom, are continuous, unbounded, and can be left skew, right skew or symmetric. With three degrees of freedom - the three prescribed percentiles - a three parameter distribution is uniquely defined. The skew normal fits this requirement (possessing shape, location and scale parameters) and the other requirements that we demand. For distributions in which the 5th and 95th percentile are symmetric around the 50th, this reduces the degrees of freedom to two, and the standard normal (Gaussian) distribution fits the bill (it is also the specific zero-skew case of the skew-normal, so is not really a different strategy).

As with many things in the fair-calibrate framework, it is a user decision - a user would be able to use different containing distributions (even non-parametric ones) if there were further information available to guide this decision. We are not sure goodness-of-fit tests are appropriate here, but if a user desired them they could be implemented.

The authors have used RMSE as a constraint to reject ensemble members. Is it possible to use correlation as a metric to select ensemble members? Different metrics target different errors, and restricting it to a single metric might lose some information.

It could, and we'll make that point. We believe RMSE is superior to correlation coefficient precisely because it encapsulates goodness of fit into a single number. The correlation coefficient would need to be used in combination with the regression slope and intercept, which would need to be close to one and zero respectively (one could get a high correlation coefficient but the wrong slope and intercept, if the pattern of internal variability between model and observations were well-matched, but too much, too little, or offsetted warming in the model.

---

## Author Response (AR2)

We thank both reviewers for their time spent in reviewing the revision. We note that reviewer #2 was satisfied with the changes made in the first revision and we trust that the documented minor revisions will further improve the manuscript.

In addition to the responses to reviewers, there is one additional change made requested by the journal. As GMD cannot reproduce coloured tables, the original shaded ranges in Table 6 have been converted to bold text. The caption has been updated and reference in the text also updated. The caption in table S4 has been updated since the shading in the supplementary material for calibration v1.4.0 has been retained.

Reviewer #1

Revision did not address my main concern, namely that paper does not show "that two very different future projections to a given emission scenario can be obtained using emissions from the IPCC Sixth Assessment Report (AR6) (fair-calibrate v1.4.0) and from updated emissions datasets through 2022 (fair-calibrate v1.4.1) for similar climate constraints in both cases."

Once again, emissions used in simulations with v1.4.1 parameters, for all scenarios except SSP2-4.5, significantly differ from those used in simulation with v1.4.0 parameters from 2015 forward (Figure 12a). Scenarios in v1.4.1 case "take into account the recent past. Scenarios used in v1.4.0 case do not. As a result difference in CO2 concentration (Fig 12b) caused not only by use of different model parameters, but also by use of different emissions
I did not suggest using the same emissions in both cases. I do understand that historical emissions were used for parameters calibration. I suggested rerunning simulation for v1.4.0 using AR6 historical plus SSP2-4.5 emissions until 2022 and scaled (harmonized) AR6 emissions from 2023. I I still think it needs to be done.

We've added a new subplot to figure 12 (12c) which shows the temperature projections from the five main SSPs under v1.4.1, v1.4.0, and v1.4.0 with historical emissions extended to 2022 under SSP2-4.5 and future projections harmonised to 2022. This neatly shows the narrowing of the plausible 21st century temperature projection space that we previously claimed (but did not demonstrate) occurs by updating the scenario start date to include 8 more years of "historical" data (either updated best estimates, in v1.4.1, or approximated by SSP2-4.5 in v1.4.0-2022-harmonized). The paragraph that relates to this plot in the main text has been duly updated.

Authors also did not explain why use of 841 ensemble member will "allows many quantiles of the full distribution to correspond to a single ensemble member at each point in time."

This minor point seems to be causing some confusion. It isn't critical to the intent of the paper, so it's cleanest to delete this part of the sentence and slightly reword. We will try and explain what we mean:

A posterior size of 841 is an author preference. A user can choose any posterior size N they like, though as stated in the paper, we recommend 500 <= N < 0.2 * #(effective samples).

841 = 840 + 1. 840 is highly composite. It exactly divides 2, 3, 4, 5, 6, 7, 8, 10, 12, 14, 15, 20, 21, 24 …; it is quite an "efficient" number in that it packs in a lot of factors into a relatively small magnitude.

Select an output variable from FaIR, let's say global mean surface temperature. Pick a timebound, say 2025. Then in 2025, you can label the 841 FaIR ensemble members from coolest to warmest in the order 0, 1, 2, …, 838, 839, 840. Let's say you want the median (50th) and 5th to 95th percentiles of the distribution. These can alternatively be thought of as the first, tenth and nineteenth vigintiles (twentieths). As 840 / 20 = 42 (exact), we can assign ensemble member 42 to be the 5th percentile, number 420 (42 * 10) to be the median, and 798 (42 * 19) to be the 95th percentile. 0 and 840 are the minimum and maximum values of the ensemble. Between every pair of adjacent vigintiles there are exactly 41 other ensemble members, and we have therefore split the ensemble into twentieths assigning a single ensemble member to each quantile without interpolation. In 2026, we can do the same, grabbing members 42, 420 and 798 as our quantiles of interest when lined up in temperature order from 0 to 840. The ensemble members corresponding to these quantiles may not be the same ensemble members that formed the percentiles in 2025. Usually, this is OK, as we are not reporting "storylines", we want to present the range of an ensemble. Because 840 is highly composite we can also do this even division into quantiles with thirds, sevenths, fifteenths, and several other numbers. Again, there is nothing special about 841 other than 840 is highly divisible; if you want hundredths, 841 won't work, but 100k + 1 will for integer k, and many earlier versions of fair-calibrate used a posterior size of 1001. It is not a requirement to create equal partitions, you can interpolate between ensemble members to get quantiles as usual.